

# Comparative actualistic study hints at origins of alleged Miocene coprolites of Poland

Tomasz Brachaniec[1], Dorota Środek[1], Dawid Surmik[1], Robert Niedźwiedzki[2], Georgios L. Georgalis[3], Bartosz J. Płachno[4], Piotr Duda[5], Alexander Lukeneder[6], Przemysław Gorzelak[7] and Mariusz A. Salamon[1]

[1] Faculty of Natural Sciences, University of Silesia in Katowice, Sosnowiec, Poland
[2] Institute of Geological Sciences of University of Wrocław, Wrocław, Poland
[3] Institute of Systematics and Evolution of Animals, Polish Academy of Sciences, Kraków, Poland
[4] Department of Plant Cytology and Embryology, Institute of Botany, Faculty of Biology, Jagiellonian University in Kraków, Kraków, Poland
[5] Faculty of Science and Technology, University of Silesia in Katowice, Sosnowiec, Poland
[6] Natural History Museum, Vienna, Austria
[7] Institute of Paleobiology, Polish Academy of Sciences, Warszawa, Poland

Corresponding author
Mariusz A. Salamon,
paleo.crinoids@poczta.fm

## ABSTRACT

Excrement-shaped ferruginous masses have been recovered from the Miocene of Turów mine in south-western Poland. These siderite masses have been the subject of much controversy, having been interpreted either as being coprolites, cololithes or pseudofossils created by mechanical deformation of plastic sediment. Here we present the results of mineralogical, geochemical, petrographic and microtomographical analyses. Our data indicate that these masses consist of siderite and iron oxide rather than phosphate, and rarely contain recognizable food residues, which may suggest abiotic origins of these structures. On the other hand, evidence in support of a fecal origin include: (i) the presence of two distinct morphotypes differing in size and shape, (ii) the presence of rare hair-like structures or coalified inclusions and (iii) the presence of rare fine striations on the surface. Importantly, comparative actualistic study of recent vertebrate feces shows overall resemblance of the first morphotype (sausage-shaped with rare coalified debris) to excrements of testudinoid turtles (Testudinoidea), whose shell fragment was found in the investigated locality. The second morphotype (rounded to oval-shaped with hair-like structures), in turn, is similar to the feces of some snakes (Serpentes), the remains of which were noted in the Miocene of the neighborhood areas. Other potential producers (such as lizards and crocodiles) and even abiotic origins cannot be fully excluded but are less likely.

## INTRODUCTION

Ferruginous masses that are excrement shaped have been recovered from the Miocene of Turów mine in south-western Poland; however, a detailed study of these masses has not been undertaken and it is unclear if they are biological or geological in origin. Examples of

the Miocene objects interpreted as coprolites are known from only a few localities in Europe, North and South America (*Amstutz, 1958*; *Roberts, 1958*; *Hunt & Lucas, 2007*, *2012*, *2021*; *Dvořák et al., 2010*; *Godfrey & Smith, 2010*; *Hunt, Lucas & Spielmann, 2012*; *Pesquero et al., 2014*; *Broughton, 2017*; *Dentzien-Dias et al., 2018*; *Tomassini et al., 2019*; *Farlow et al., 2020*). Giant earthworms, fish, rodents, notoungulates, hathliacynid and borhyaenoid marsupials, indeterminate carnivorans, sirenians, crocodilians, were commonly invoked as potential producers of these coprolites (*Broughton, Simpson & Whitaker, 1977*, *1978*; *Godfrey & Smith, 2010*; *Broughton, 2017*; *Dentzien-Dias et al., 2018*; *Tomassini et al., 2019*). The majority of described Miocene vertebrate coprolites were produced by carnivores. This is not surprising because calcium phoshate derived from undigested bones in the feces of carnivores acts as important source of perminalizing agent which is often not present in the feces of herbivorous tetrapods (*e.g.*, *Hunt, Chin & Lockley, 1994*; *Pesquero et al., 2011*; *Dentzien-Dias et al., 2018*). Excrement-shaped ferruginous masses have been considered (based on morphological grounds) by some authors as being coprolites (*Amstutz, 1958*; *Broughton, Simpson & Whitaker, 1977*, *1978*) or cololites (*Seilacher et al., 2001*; *Broughton, 2017*; intestinal casts—evisceralites; see *e.g.*, *Hunt & Lucas, 2021*). Notably, *Broughton (2017)* has recently described several dozen centimeters long intestine-like elongated objects revealing bilateral symmetry in cross-section and surface features consisting of fine longitudinal parallel striations, which were ascribed to gut casts of a previously unrecognized giant terrestrial earthworm.

Excrement-shaped masses are commonly reported from clay-rich sediments ranging in age from Permian to Holocene. However, given their ferruginous composition, significant variation in size, lack of internal inclusions, and scarcity of associated vertebrate remains, most authors rejected a zoological origin (*Dake, 1939*, *1960*; *Danner, 1968*, *1994*, *1997*; *Major, 1939*; *Roberts, 1958*; *Spencer & Tuttle, 1980*; *Love & Boyd, 1991*; *Spencer, 1993*, *1997*; *Hardie, 1994*; *Mustoe, 2001*). Different non-zoological hypotheses have been invoked to explain the origins of these objects (such as soft sediment extrusion triggered by coseismic liquefaction (*Peterson & Madin, 1997*)), sediment intrusion into hollow logs (*Spencer & Tuttle, 1980*; *Spencer, 1993*) or squeezed between plant stems (*Roberts, 1958*), expulsion of sediment in response to gravity (*Love & Boyd, 1991*), and extrusion of siderite related to methanogenesis (*Love & Boyd, 1991*; *Mustoe, 2001*).

Until recently, a detailed study on excrement-shaped ferruginous masses from the Miocene of Poland has been lacking. In this paper we analyse the Miocene excrement-shaped specimens collected from the coal mine of Turów for the first time. According to our results, we favour the hypothesis that the specimens from Poland may represent true coprolites and more particularly pertaining to two different reptile groups.

## Geological setting

The Turów lignite mine is located in the south-western part of the Lower Silesia Voivodeship (SW Poland) and covers the former village of Turów. It is located in the central part of the mesoregion Żytawa-Zgorzelec Depression located between the state borders of Germany and the Czech Republic (Fig. 1A). Turów lignite deposits are part of the Upper Lusatian Brown Coal Basin. This basin comprises a few tectonic sinkholes

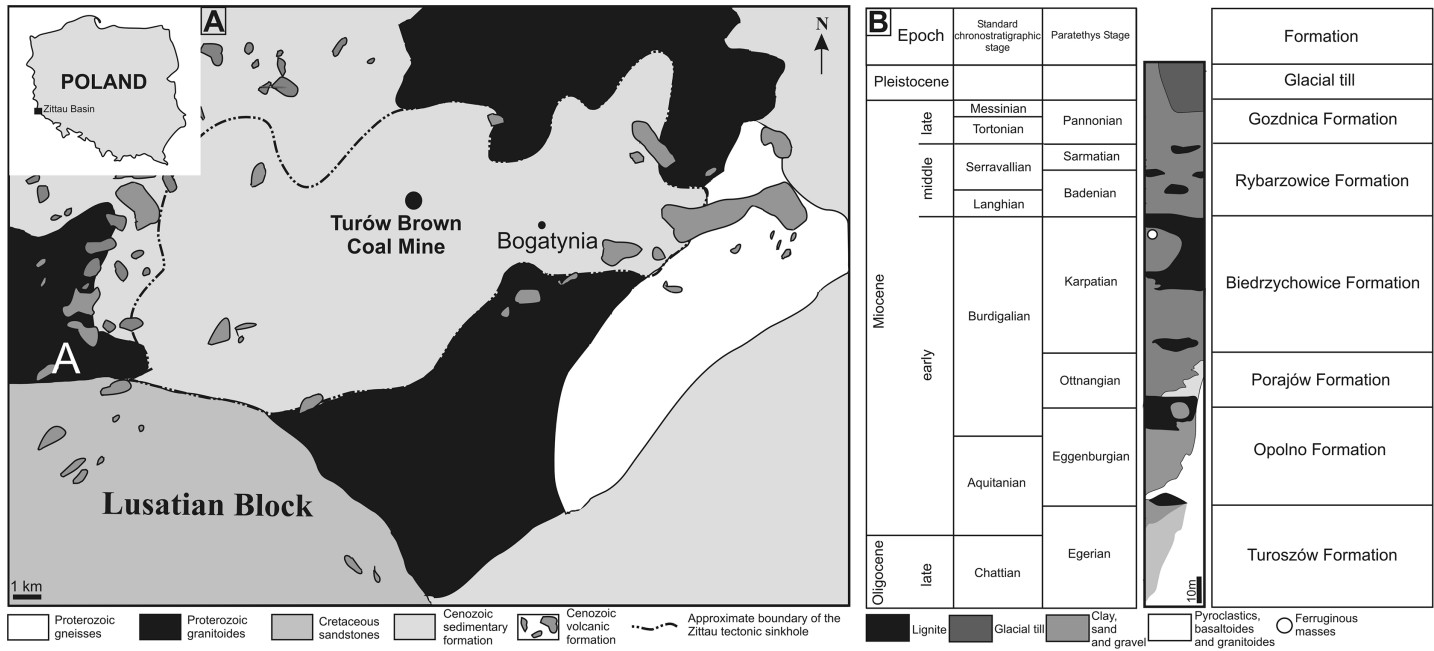

**Figure 1** (A) Map of Poland and Zittau Basin. (B) Simplified lithostratigraphic section of Paleogene and Neogene sediments of the Polish part of the Zittau Basin (simplified and modified after *Kasiński et al., 2015*; *Paszcza, 2021*).

(*Piątkowska, Kasiński & Graniczny, 2000*) that developed in Paleogene at the junction of two regional zones of strong tectonic activity: Ore Mts. Graben (Ohrza rift) and the Lusatian-Elbe Tectono-Volcanic Zone (*Jęczmyk & Sztromwasser, 1998*). The most southern of these is the Zittau basin (Figs. 1A, 1B), which was filled mainly by limno-fluvial or limnic clays, silts, sands and thick layers of lignites exploited in the Turów mine (*Kasiński, 2000*; *Kasiński et al., 2015*). Furthermore, there are numerous vulcanite rocks of late Eocene, Oligocene and early Miocene age (*Kasiński et al., 2015*). The basal part of sedimentary section of the Zittau basin is not older than the early Oligocene, however most sediments were formed in the Miocene (*Kasiński et al., 2015*). At the base of the Zittau Basin, Precambrian and Palaeozoic metamorphic and igneous rocks are present (*Marcinkowski, 1985*).

The lithological profile of the Turów mine is ca. 250 m thick and consists of seven lithostratigraphic units of sedimentary rocks. Apart from the two youngest units (Gozdnica Fm. and glacial tills), all of them are mainly composed of clays and/or muds with intercalations of sands or sands with gravels. Additionally, there are coal seams, especially in Opolno and Biedrzychowice formations, which are the deposits mined at the Turów mine. The oldest Cenozoic sediments of the profile are Oligocene sediments (Egger age), forming the lower and middle part of the Turoszów Fm. (*Kasiński et al., 2015*). The youngest in the profile are sands and gravels of the Gozdnica Fm. and Pleistocene deposits, mainly represented by tills (*Kasiński et al., 2015*).

The Turoszów Fm. was formed in fluvial and limnofluvial conditions, while the Opolno and Biedrzychowice Fms have been formed in limnotelmatic environments (sediments formed in the terrestrial environment most often as a result of sea regression), and Porajów

Fm. represents limnofluvial environment. Sediments of the upper part of Miocene profile (Rybarzowice and Gozdnica Fms) have alluvial origin (*Kasiński, Badura & Przybylski, 2003*). Biedrzychowice Fm., within which excrement-shaped ferruginous masses and a turtle remain have been documented, was formed in vast swamps and rushmarshes. However, palaeobotanical data indicate that during the deposition of the upper part of this formation terrestrial forests (mezophilus mixed forests and upland mixed forests) also grew locally (*Kasiński et al., 2010*). Especially in the upper part of this formation, there are numerous palaeosols levels with plant roots and trunks preserved *in situ* (*Kasiński & Wiśniewski, 2003*). Intercalations of sands and sands with gravels, representing a braided river environment, are also present (*Kasiński et al., 2010*). Marsh forests mainly composed of *Cupressaceae* and *Taxaceae* (*Sadowska, 1995*; *Kasiński & Wiśniewski, 2003*). Based on palaeobotanical analysis of the coal seam it was concluded that there was a humid warm temperate climate similar to that of south-eastern China today (*Durska, 2008*; *Kasiński et al., 2010*). The excrement-shaped ferruginous masses and the turtle shell fragment documented in this article (Figs. 2A, 2B, 2E, 2G, 2N) were collected in an inactive part of the excavation, at the top layer of clay complex, which is covered by quartz sand layer of the highest part of Biedrzychowice Fm. in the uppermost part of early Miocene (Burdigalian; see Fig. 1C). Two distinct morphotypes randomly distributed within clay and mudstone on the flat surface of the excavation were noted.

## Fossil content in the Turów area and adjacent areas

No animal fossil remains have been documented so far from the Oligocene–Miocene of the Zittau Basin with exception of burrows in sediment produced by indeterminate sediment eating fauna (*Kasiński et al., 2015*). Apart from the excrement-shaped ferruginous masses, a fragment of a turtle shell was found in the Biedrzychowice Fm. This shell fragment (see Fig. 2N) can only be identified as an indeterminate testudinoid. This turtle lineage is otherwise abundant in Oligocene and early Miocene localities in Germany and Czech Republic (*von Reinach, 1900*; *Młynarski & Roček, 1985*), but had not so far been documented from coeval localities in Poland. In older, Eocene and Oligocene localities in the neighbouring north-western Czech Republic and south-eastern Germany (Saxony and southeastern Saxony-Anhalt), rich assemblages of terrestrial-aquatic tetrapod fauna have been documented, comprising frogs, salamanders, choristoderans, crocodiles (also crocodile coprolites, see *Kasiński et al. (2015)*, and literature cited therein), turtles, lizards, and snakes (Table 1).

On the other hand, in the lower Miocene clays and sands of North Bohemian Brown Coal Basin, a very rich fauna assemblage was reported (*Klembara, 1979*, *1981*; *Roček, 1984*; *Szyndlar, 1991a*, *1991b*; *Szyndlar & Schleich, 1993*; *Szyndlar & Rage, 2003*; *Čerňanský, 2010a*, *2010b*; *Dvořák et al., 2010*). The latter mentioned and illustrated numerous vertebrates represented by osteichthyan fish, amphibians, reptiles, birds, and mammals. The reptile taxa are shown in the table below (Table 2).

Brachaniec et al.
2022
10.7717/peerj.13652

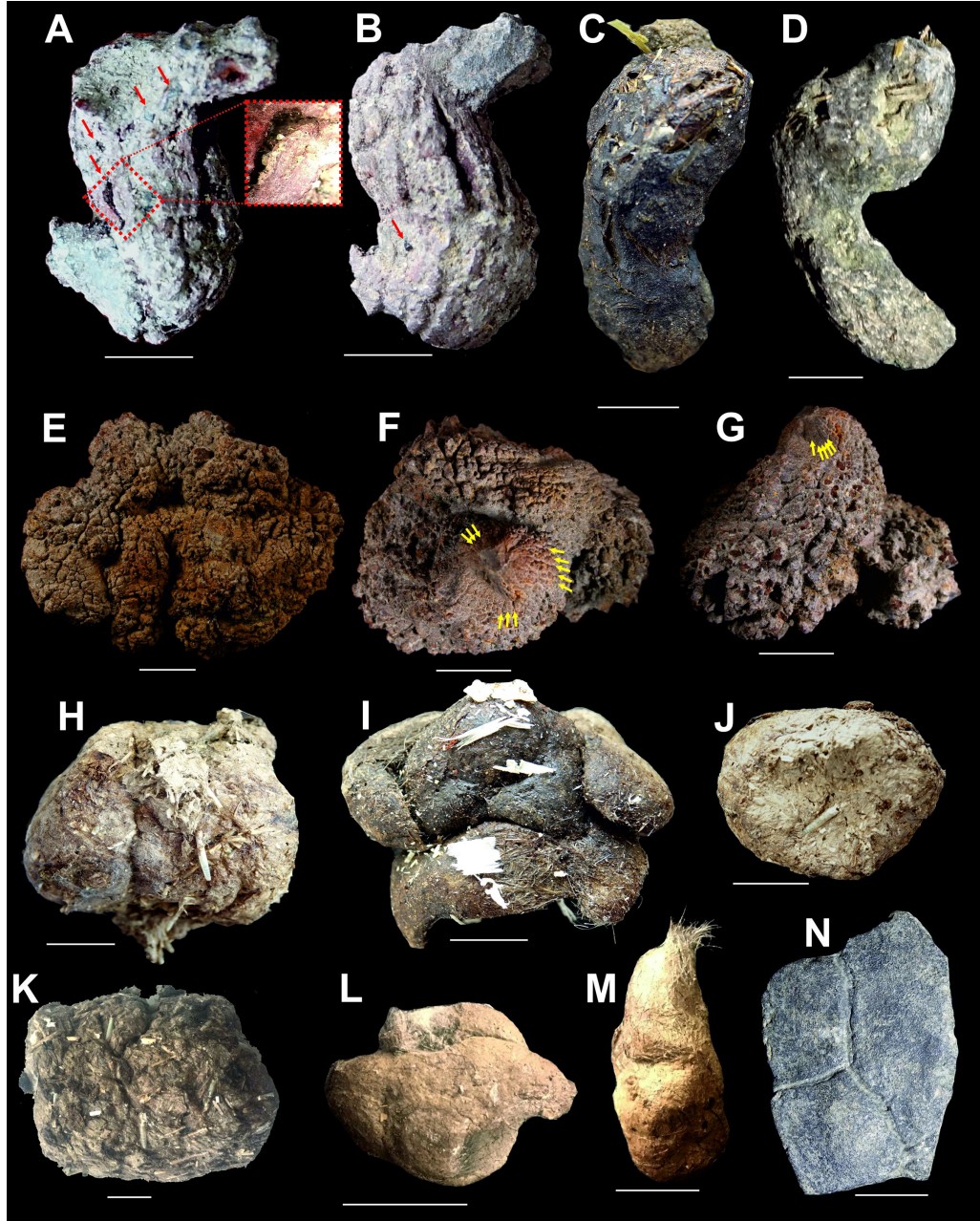

**Figure 2** Miocene excrement-shaped ferruginous masses from the Turów lignite mine, Poland interpreted as turtle and snake coprolites (A, B, E–G), compared with modern turtle and snake excrements (C, D, H–M) and fossil remain from the Turów lignite mine, Poland (N). Scale bar equals 10 mm. (A, B) coprolites, morphotype M1. (C, D) modern excrements of *Testudo horsfieldii* (C), and *Testudo hermanni* (D). (E–G) coprolites, morphotype M2. (H, I) modern excrements of *Python regius*. (J) modern excrement of *Boa constrictor*. (K) modern excrement of *Ophiophagus hannah*. (L) modern excrement of *Elaphe anomala*. (M) modern excrement of *Vipera berus*. (N) Shell fragment of Testudinoidea indet. from the Turów lignite mine, Poland. Red arrows in A and B = coalified inclusions, yellow arrows in F and G = fine striations.

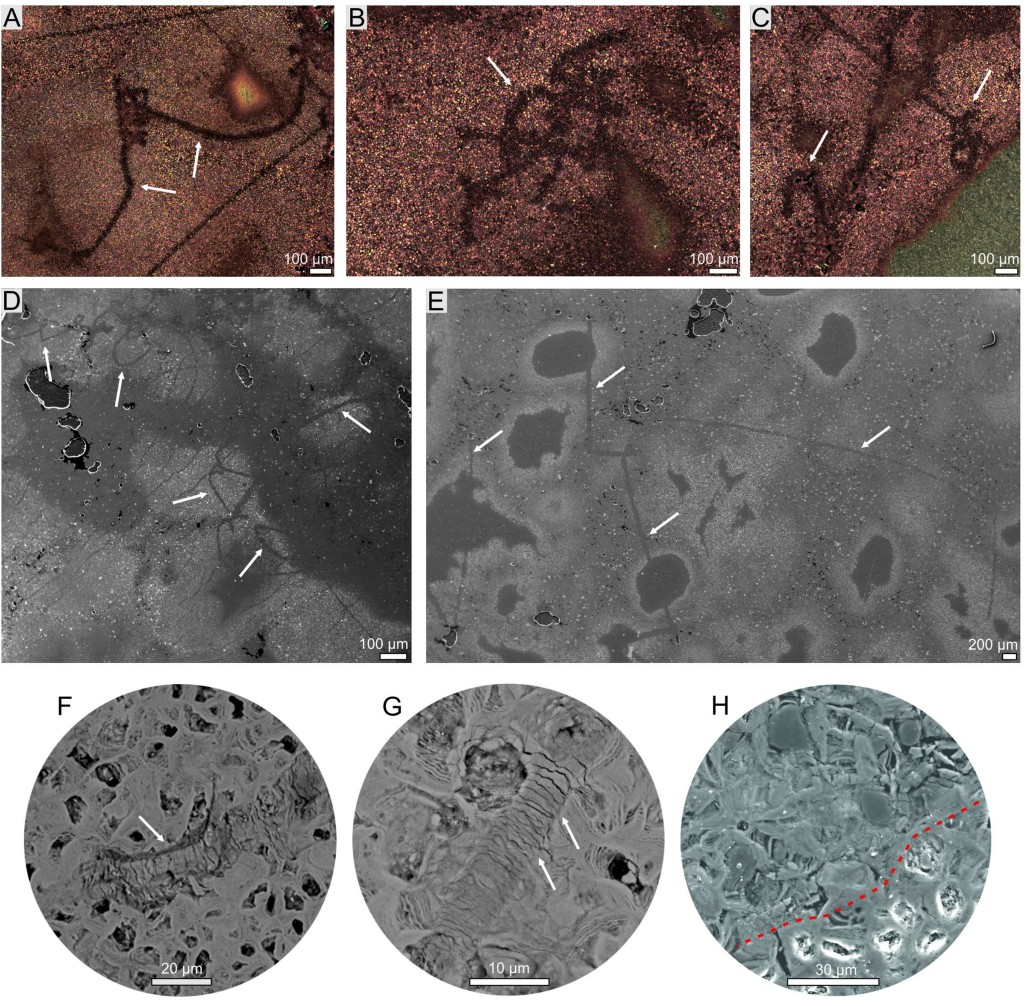

**Figure 3 Hair-like structure identified in coprolites (morphotype 2).** (A–C) Optical microscopy. (D–H) SEM images. (F, G) Magnification of scale-like pattern. (H) Magnification of internal hair-like cellular structure (above red dotted line).

## MATERIALS AND METHODS

Among 29 ferruginous masses obtained from the investigated locality, 10 representative specimens were selected for detailed investigation. All specimens are housed in the Institute of Earth Sciences of the University of Silesia in Katowice, Poland, and catalogued under registration number GIUS 10-3739.

A clay sample weighing ca. 40 kg was also collected from Biedrzychowice Fm. and transported to the Laboratory of the Institute of Earth Sciences of the University of Silesia in Katowice. It was washed using running hot tap water, screened on a sieve column (Ø1.0, 0.315 and 0.1 mm-mesh respectively), and finally dried at 180°. This washed and dried residue was observed under a Leica WildM10 microscope for vertebrate microremains; no fossils were found in the residue.

Ferruginous masses recorded herein have been investigated with a number of different analytical tools.

**Table 1** Oligocene amphibians and reptiles collected in adjacent areas (north-western Czech Republic and south-eastern Germany (Saxony, south-eastern Saxony-Anhalt)); after *Laube, 1901*; *Obrhelová, 1971*; *Špinar, 1972*; *Obrhelová & Obrhel, 1987*; *Szyndlar, 1994*; *Böhme, 1996, 1998, 2007*; *Gaudant, 1996, 1997*; *Micklich & Böhme, 1997*; *Bellon et al., 1998*; *Kvaček & Walther, 2003*; *Mikuláš et al., 2003*; *Fejfar & Kaiser, 2005*; *Karl, 2007*; *Čerňanský & Augé, 2012, 2013*; *Čerňanský, Klembara & Müller, 2016*; *Georgalis & Joyce, 2017*; *Chroust, Mazuch & Luján, 2019*.

| Age | Locality | Amphibians | Reptiles |
|---|---|---|---|
| Late Oligocene | Lužice-Žichov (Czech Republic) | *Triturus opalinus* *Rana luschitzana* *Asphaerion reussi* | |
| | Suletice (Czech Republic) | *Archaeotriton basalticus* *Palaeobatrachus grandipes* *Palaeobatrachus laubei* | |
| | Bechlejovice (Czech Republic) | *Archaeotriton basalticus* *Palaeobatrachus diluvianus* *Palaeobatrachus luedeckei* *Palaeobatrachus robustus* *Palaeobatrachus grandipes* *Palaeobatrachus novotnyi* *Eopelobates bayeri* | '*Diplocynodon*' sp. |
| Early Oligocene | Espenhain, Saxony (Germany) | | Trionychidae indet. *Pelorochelon* sp. *Diplocynodon* sp. |
| | Kundratice (Czech Republic) | *Palaeobatrachus* sp. cf. *Eopelobates* sp. | cf. *Diplocynodon* sp. |
| | Lukavice (Czech Republic) | | '*Diplocynodon*' sp. |
| | Markvartice (Czech Republic) | *Chelotriton laticeps* *Palaeobatrachus diluvianus* *Palaeobatrachus luedeckei* *Palaeobatrachus* sp. | |
| | Dětaň (Czech Republic) | Salamandridae indet. Palaeobatrachidae indet. Pelobatidae indet. Discoglossidae indet. | Lacertidae indet. Anguidae indet. Testudinidae indet. Serpentes indet. Crocodylia indet. |

## Optical microscopy and thin-sectioning

Optical observation of several thin sections from four ferruginous masses representing two morphotypes, and two representative concretions were carried out using Leica SZ-630T dissecting microscope and Nikon Eclipse E100 light microscopy, while the microphotographs were collected using Olympus BX51 polarizing microscope equipped with an Olympus SC30 camera and a halogen light source, installed Faculty of Natural Sciences at the University of Silesia in Katowice (Sosnowiec, Poland) (Figs. 3, 4).

Thin sections were made in the Grindery at the Faculty of Natural Sciences, University of Silesia in Katowice, Sosnowiec, Poland. Specimens were embedded in Araldite epoxy resin, sectioned, mounted on the microscope slides and polished with silicon carbide and aluminium oxide powders to about 30 μm thick.

**Table 2 Reptiles recorded in the lower Miocene deposits of North Bohemia, Czech Republic (taken from *Klembara, 1981, 2008, 2012, 2015; Młynarski & Roček, 1985; Ivanov, 2002; Čerňanský & Joniak, 2009; Čerňanský, 2010a, 2010b, 2012; Čerňanský & Bauer, 2010; Čerňanský, Rage & Klembara, 2015; Dvořák et al., 2010; Joyce, 2016; Georgalis & Joyce, 2017; Klembara & Rummel, 2018; Chroust et al., 2021*).**

| Turtles | Crocodiles | Lizards | Snakes | Choristoderans |
|---|---|---|---|---|
| | | | Scolecophidia indet. | |
| | | | *Bavarioboa hermi* | |
| | *Diplocynodon* cf. *ratelii* | *Merkurosaurus ornatus* | *Bavarioboa* sp. | *Lazarussuchus dvoraki* |
| | | | Constrictores indet. | |
| | | | *Falseryx petersbuchi* | |
| | | *Pseudopus ahnikoviensis* | "*Coluber*" *dolnicensis* | |
| | | *Pseudopus confertus* | *Texasophis bohemiacus* | |
| | | *Pseudopus* sp. | | |
| | | *Ophisaurus fejfari* | | |
| *Rafetus bohemicus* | | *Ophisaurus holeci* | "*Coluber*" *suevicus* | |
| | | *Ophisaurus robustus* | | |
| | | *Ophisaurus spinari* | | |
| | | *Ophisaurus* aff. *spinari* | | |
| | | *Ophisaurus* sp. (two morphotypes) | | |
| | | Anguinae indet. (several morphotypes) | | |
| | | *Palaeocordylus bohemicus* | | |
| Trionychinae indet. | | aff. *Palaeocordylus bohemicus* | "*Coluber*" *caspioides* | |
| | | | "Colubrinae" indet. | |
| *Ptychogaster laubei* | | *Euleptes gallica* | *Elaphe* sp. | |
| *Ptychogaster* cf. *emydoides* | | *Chamaeleo andrusovi* | | |
| *Ptychogaster* sp. | | Chamaeleonidae indet. | *Natrix sansaniensis* | |
| Testudinidae indet. | | *Amblyolacerta dolnicensis* | | |
| *Chelydropsis* sp. | | *Lacerta* sp. | *Natrix merkurensis* | |
| | | *Miolacerta tenuis* | *Neonatrix nova* | |
| | | Lacertidae indet. | *Palaeonatrix lehmani* | |
| | | cf. Scincidae indet. | | |
| | | *Blanus gracilis* | Natricinae indet. | |
| | | Squamata indet. | Elapidae indet. | |
| | | | *Macrovipera platyspondyla* | |
| | | | *Vipera antiqua* | |
| | | | *Vipera* sp. | |
| | | | Viperidae indet. | |

## Scanning electron microscopy and microtomography

The chemical composition, morphology of ferruginous masses matrix and microstructures topography were investigated using the desktop scanning electron microscopy (SEM) Phenom XL, Phenom World (Thermo Fisher Scientific, Eindhoven, Netherlands) equipped with a fully integrated energy-dispersive X-ray spectroscopy (EDS) detector and secondary electron detector (SED) located in the Faculty of Natural Sciences at the

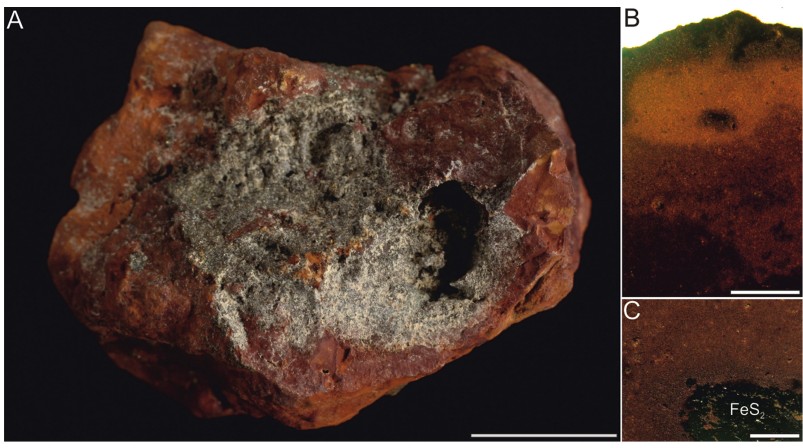

**Figure 4 Typical concretion from the Turów lignite mine, Poland. (A) General view. (B, C) Magnification of internal structure. Scale bar equals 1 cm (A) and 0.5 mm (B, C).**

University of Silesia in Katowice (Bankowa, Katowice, Poland) (Fig. 5). Measurements were performed with low-vacuum settings with accelerating voltage 15 kV. Virtual sections of a selected specimen (GIUS 10-3739/23; Movie S1) were made in the Faculty X-ray Microtomography Laboratory at Faculty of Computer Science and Material Science, University of Silesia in Katowice, Chorzów, Poland using the General Electric Phoenix v| tome|x micro-CT equipment at 160 kV, 70 μA and scanning time of 20 min. Projection images were captured using a 1,000 × 2,024 pxs scintillator/CCD with an exposure time of 250 ms and processed using Volume Graphics® VGSTUDIO Max software and analysed using Volume Graphics® myVGL viewer.

## XRD and confocal Raman spectroscopy

Bulk mineral composition of two powdered specimens representing each morphotypes was determined by Debye–Sherrer X-ray method using Rigaku SmartLab diffractometer equipped with Cu Kα1 source radiation. Measurement parameters were: acceleration voltage: 45 kV; filament current: 200 mA; step size: 0.05° 2Θ. Analyses of the collected data were carried out by means of XRAYAN Software using the newest ICSD database (Fig. 6).

To determine the more detailed mineralogical composition, the WITec confocal Raman microscope CRM alpha 300M equipped with an air-cooled solid state laser (λ = 532 nm and λ = 457 nm) and an electron multiplying CCD (EMCCD) detector was used. The calibration of the instrument was verified by checking the Si position. The Raman scattered light was focused onto a multi-mode fiber and monochromator with a 600 line/mm grating. To collect spectra of the coprolite matrix phases, the 50×/0.76 NA and 100×/0.9 NA air Olympus MPLAN objectives were used. All spectra were collected in the 200–4000 cm$^{-1}$ range with 3 cm$^{-1}$ spectral resolution (Figs. 7, 8). A surface Raman imaging map was collected in a 140 × 25 μm area using 140 × 20 pixels with an integration time of 0.5 s per spectrum, and precision of moving the sample during the measurements of ±0.5 μm. The cluster analysis was performed to group spectra into clusters. K-means analysis with the Manhattan distance for Raman imaging maps was carried out. The data

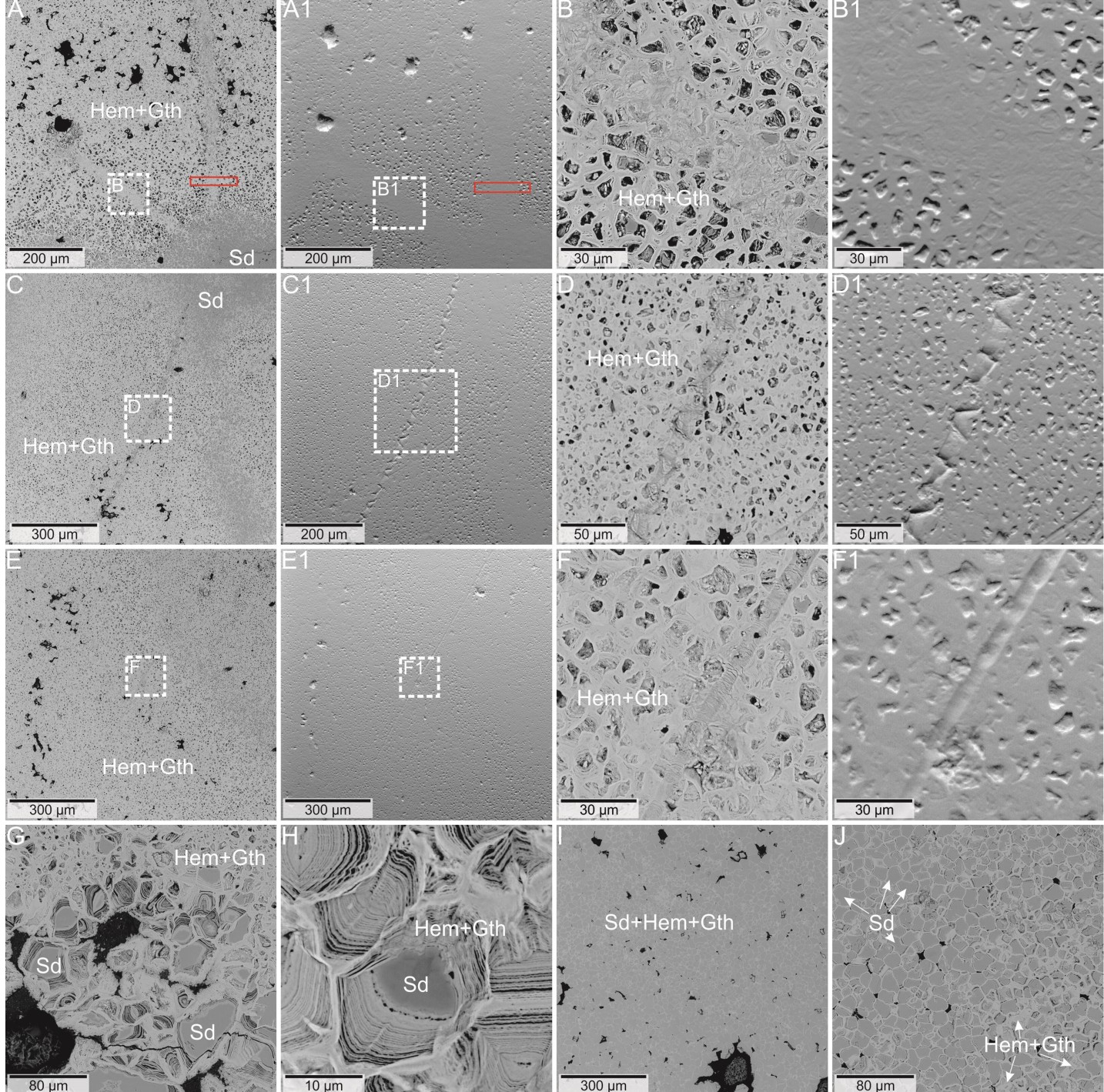

**Figure 5** **(A–F) BSE image of the coprolite matrix with preserved structures. The red frame indicates the area of the Raman image from Fig. 5.** (A1–F1) topographic pictures of the area from the A–F images. (G, H) BSE image of multi-layered iron oxides form with siderite center. (I, J) BSE image of the non-porous type of coprolite matrix. Mineral abbreviations: Gth - goethite, Hem - hematite, Sd - siderite.

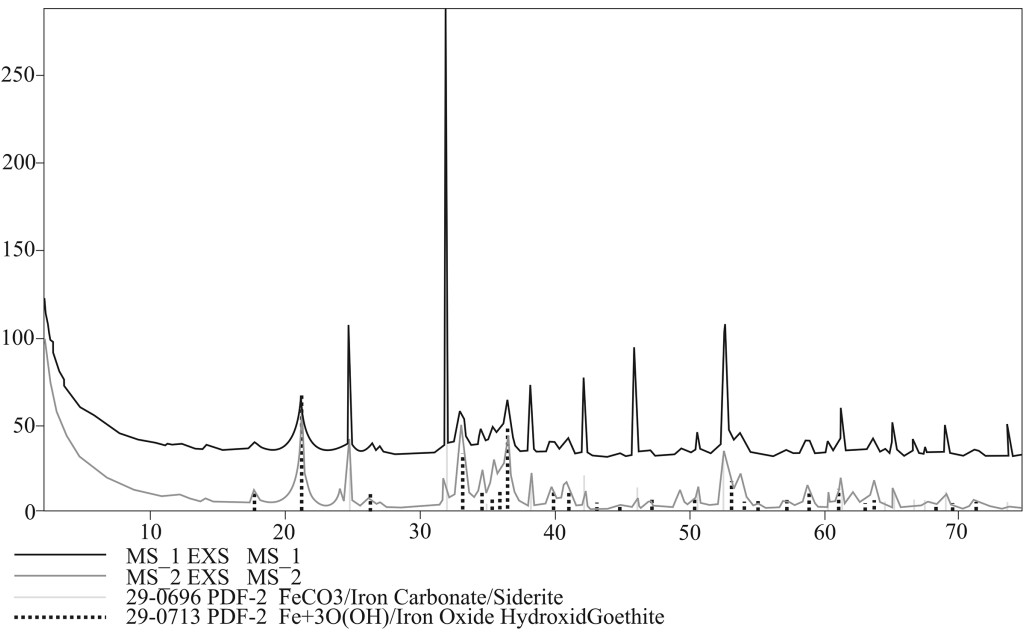

**Figure 6 XRD diffractograms for two coprolite morphotypes. Black line, morphotype 1; dark grey line, morphotype 2.**

obtained was manipulated by WITec Project FIVE Software (cosmic rays removal procedure and cluster analysis) and GRAMS software package (baseline correction).

For comparison purpose, a bulk sample from concretion was also analyzed (Fig. 9). XRD scan was performed using a PANalytical X'Pert Pro X-ray diffractometer (PW3040/60), equipped with a cobalt anode tube (40 mA, 40 kV) and an X'Celerator detector. The analyses were performed over a wide range of 2Θ (from 5 to 90) to obtain the appropriate number of peaks for each phase, with a long counting time constant (300 s) and precise step size (0.02°2Θ). The obtained results (Fig. 9) were evaluated in the HighScore program, version 4.9 (*Degen et al., 2014*) linked to the ICSD (ver. 2015), COD (ver. 2021), and ICDD PDF4 + databases (ver. 2019).

## Observations of extant excrements

External and internal features of feces of Recent animals (private farms and from wild animals from local forests or raised at the Silesian Zoological Garden in Chorzów, Poland) were analyzed (Figs. 2, 10). Over the course of 2 months, ca. 800 excrements belonging to modern fish, amphibians, reptiles, birds and mammals were collected (a detailed study on these feces is planned to be published elsewhere in the future). Lineages that had their representatives in the early Miocene sediments of North Bohemia, Czech Republic (see Table 3) were selected for a more detailed observation. Additionally, the animals had to be large enough to produce excrements with dimensions comparable to those currently documented in the fossil state. Thus, the feces of small fish, the remains of which are known from the Miocene sediments of North Bohemia, such as *Chalcaburnus* or *Nemacheilus*, toads and frogs (*Bufo*, *Rana*, *Pelobates*), birds (*Upupa*, *Coturnix*) and mammals (Chiroptera, *Dryomys*, *Sciurus*, *Martes*), were not taken into account.

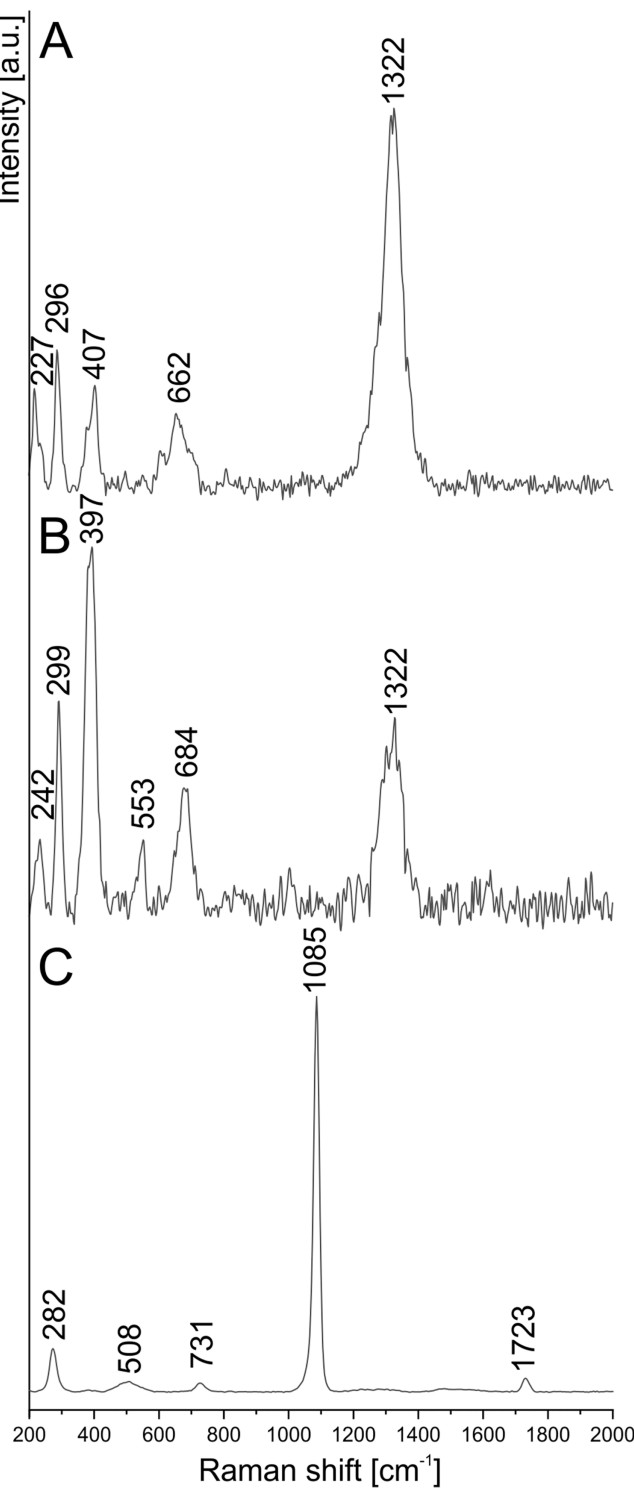

**Figure 7 Raman spectrum of (A) hematite, (B) goethite, (C) siderite from coprolite matrix.**

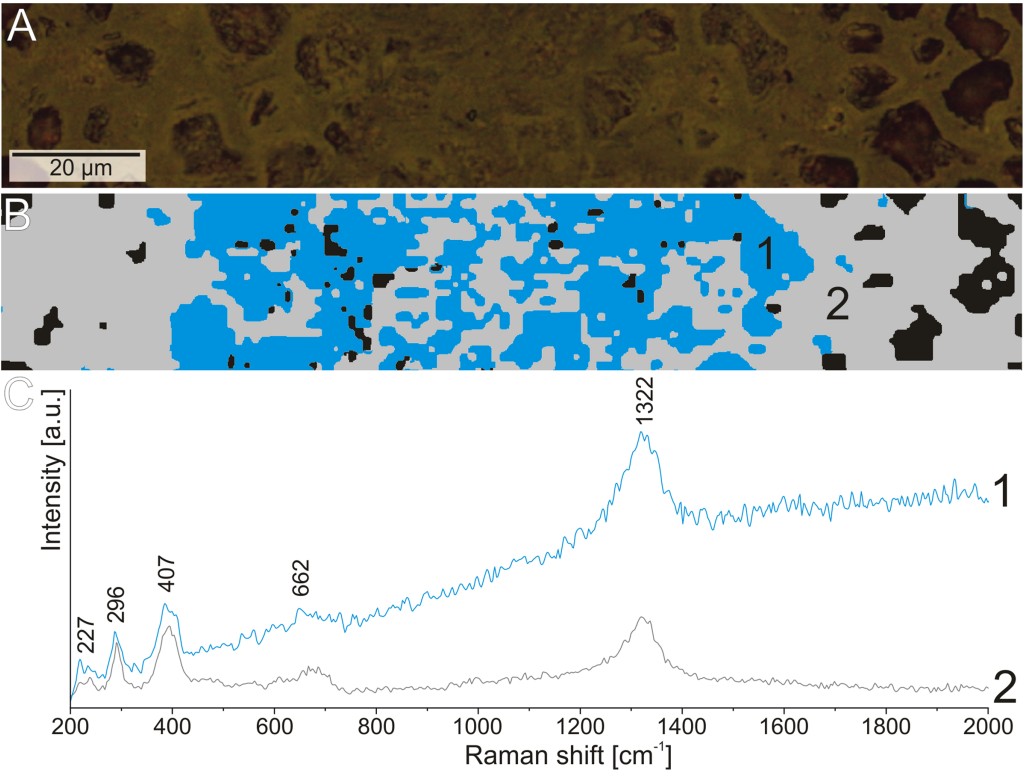

**Figure 8 The difference in the fluorescence level in the Raman spectrum of hematite from the structures.** (A) Reflected light image of the elongated structure. (B) Cluster analysis of the structure and matrix Raman mapping. (C) Raman spectrum of hematite from structure area (1) and coprolite matrix (2).   

   A special attention was also paid to those animals that left at least part of their feces in the terrestrial environment (see Figs. 2H–2M, 10). These were snakes (king python (*Python regius*), tiger python (*Python molurus*), reticulated python (*Malayopython reticulatus*), common boa (*Boa constrictor*), king cobra (*Ophiophagus hannah*), Korean rat snake (*Elaphe anomala*), common European viper (*Vipera berus*)), lizards (komodo dragon (*Varanus komodoensis*)), and turtles (Mediterranean tortoise (*Testudo hermanni*), steppe tortoise (*Testudo horsfieldii*), Indian star tortoise (*Geochelone elegans*), Spanish pond turtle (*Mauremys leprosa*), and Nile soft shell turtle (*Trionyx triunguis*)). Excrements of the European beaver (*Castor fiber*), African lion (*Panthera leo*), cheetah (*Acinonyx*), Indian star tortoise (*Geochelone elegans*), and white-tailed Eagle (*Haliaeetus albicilla*) were additionally analyzed (Fig. 10).

## RESULTS

### General morphology

A total of 29 specimens of excrement-shaped ferruginous masses were collected. Among these specimens two different shapes and sizes were identified. More specifically, morphotype 1 (M1) is represented by small (mean length: 33.4 mm, see Table 4) sausage-shaped (=allantoid) specimens with smooth or rough surface and flared lower part

**Table 3 List of taxa of observed vertebrate excrements in zoos and private farms. Fossil representatives taken from *Hunt & Lucas (2007)*.**

| Modern fish | Fossil representative |
|---|---|
| *Aspius* sp. | *Aspius* sp., *Aspius laubei*, *Barbus* sp., *Barbus bohemicus* |
| *Leuciscus* sp. | *Palaeoleuciscus* sp., *Palaeoleuciscus chartacerus* |
| *Tinca* sp. | *Palaeotinca* sp., *Palaeotinca egeriana*, *Palaeotinca obtruncata* |
| **Modern amphibians** | **Fossil representative** |
| *Andrias* sp. | *Andrias bohemicus*, *Andrias* cf. *scheuchzeri*, *Chelotriton* cf. *paradoxus* |
| **Modern birds** | **Fossil representative** |
| *Phalacrocorax* sp. | *Nectornis* sp., *Phalacrocorax littoraris* |
| Strigiformes indet. | *Prosybris antiqua*, *Mioglaux debellatrix*, *Intulula tinnipara* |
| *Aquila* sp. | *Polemaetus* sp. |
| *Accipiter* sp. | Accipitridae indet. |
| **Modern mammals** | **Fossil representative** |
| *Ursus* sp. | *Ursavus elmensis*, *Ursavus isorei*, *Tomocyon* sp., *Hemicyon* cf. *stehlini*, *Amphicyon bohemicus*, *Amphicyon major*, *Megamphicyon giganteus*, *Cynelos schlosseri* |
| Rhinocerotidae indet. | *Mesaceratherium* aff. *paulhiacense*, *Prosantorhinus laubei*, *Protaceratherium minutum* |
| *Cervus* sp. | *Procervulus* cf. *praelucidus* |
| Equidae indet. | *Anchitherium aurelianense* |

(Figs. 2A, 2B). Morphotype 2 (M2), in turn, is represented by larger (mean length: 46.3 mm) and more rounded to oval, massive specimens with rough surface (Figs. 2E–2G). Rare specimens (Figs. 2F, 2G) included into M2, bear prominent pointed end covered by striate pattern (herein interpreted as a trace produced after closing anus, see Discussion below). Color of both morphotypes varies from pale orange, through greenish red, to burgundy-colored.

## Optical microscopy, microtomographic and palaeontological studies

The thin sections from two specimens representing two morphotypes were studied both under transmitted and reflected light optical microscopy. The sections from both morphotypes look very similar. They are dominated by darker matrix almost not translucent making transmitting light observations difficult. The mineral matrix seems to be rather homogenous. Within the matrix of M2-type more translucent elongated straight or curly structures (up to about few mm long and 10–99 μm thick; mean: 52 μm) are visible (Fig. 3). The structures sometimes form arcs or are twisted. In the reflected light, they seem to be areas of light reduction, while surrounding matrix is oxidized. The dark (rusty-colored, brown to almost black), poorly translucent coloring of a matrix suggests iron-rich mineral which form the matrix. Therefore both mineral matrix as well as thin elongated straight to curly structures were studied in-depth under SEM and Raman imaging (see below). No other distinguishable microremains were noticed in thin

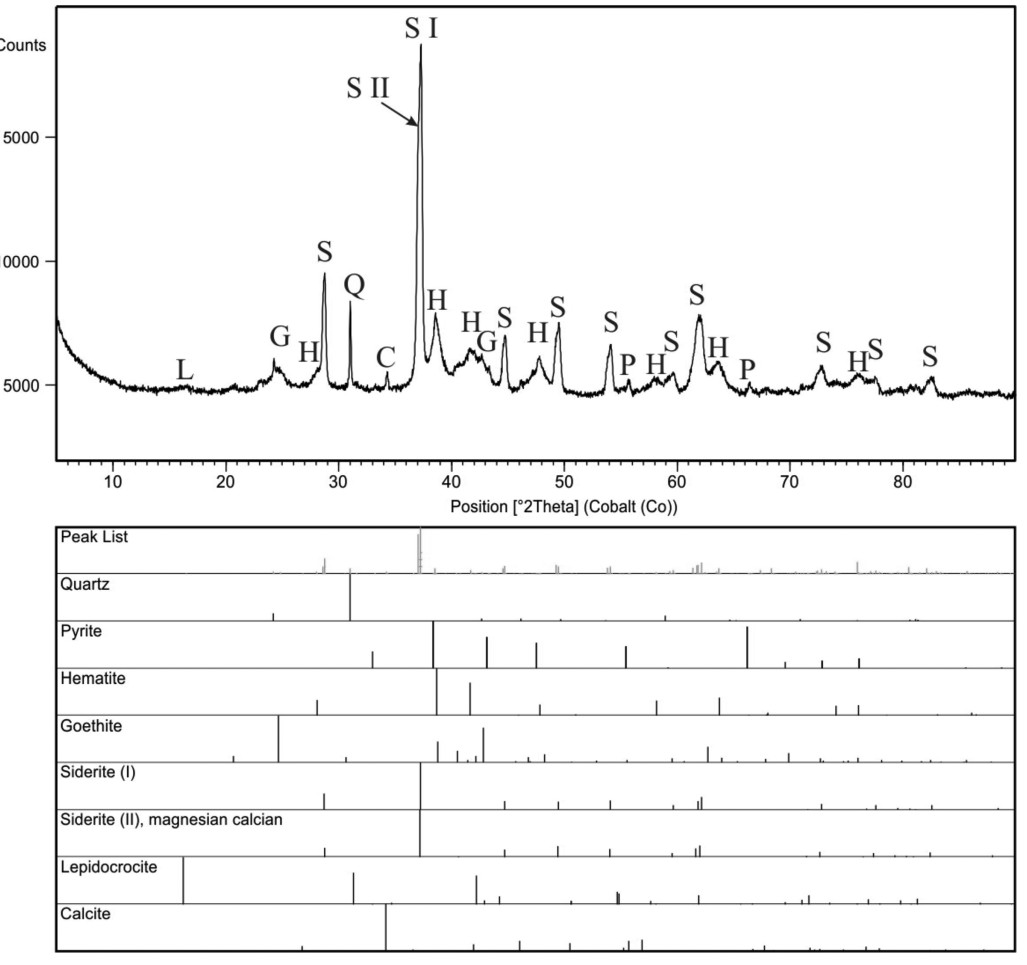

**Figure 9 X-ray diffraction pattern of the concretion sample, mineral phases are marked with symbols: C, calcite; G, goethite; H, hematite; L, lepidocrocite; P, pyrite; Q, quartz; S, siderite.** Two generations of siderite are additionally marked as S I (pure siderite) and S II (magnesian, calcian siderite).                

sections. However, at the broken surfaces of some specimens of the first morphotype (M1) some tiny coalified debris were occasionally noticed (Figs. 2A, 2B).

Microtomographical studies of selected specimen (GIUS 10-3739/23; Movie S1) did not reveal any internal structures, which could have been eventually interpreted as undigested food remains.

The external appearance and internal structure of concretions from Turów mine are markedly different (Fig. 4). They are red to brown and irregular in outline. Locally they display a zonal structure and contain well-visible pyrite crystals.

## Mineralogical, geochemical and structural analyses

XRD analyses of powdered fragments of two specimens from each morphotypes indicated that they both are composed of siderite with small admixture of geothite, and hematite.

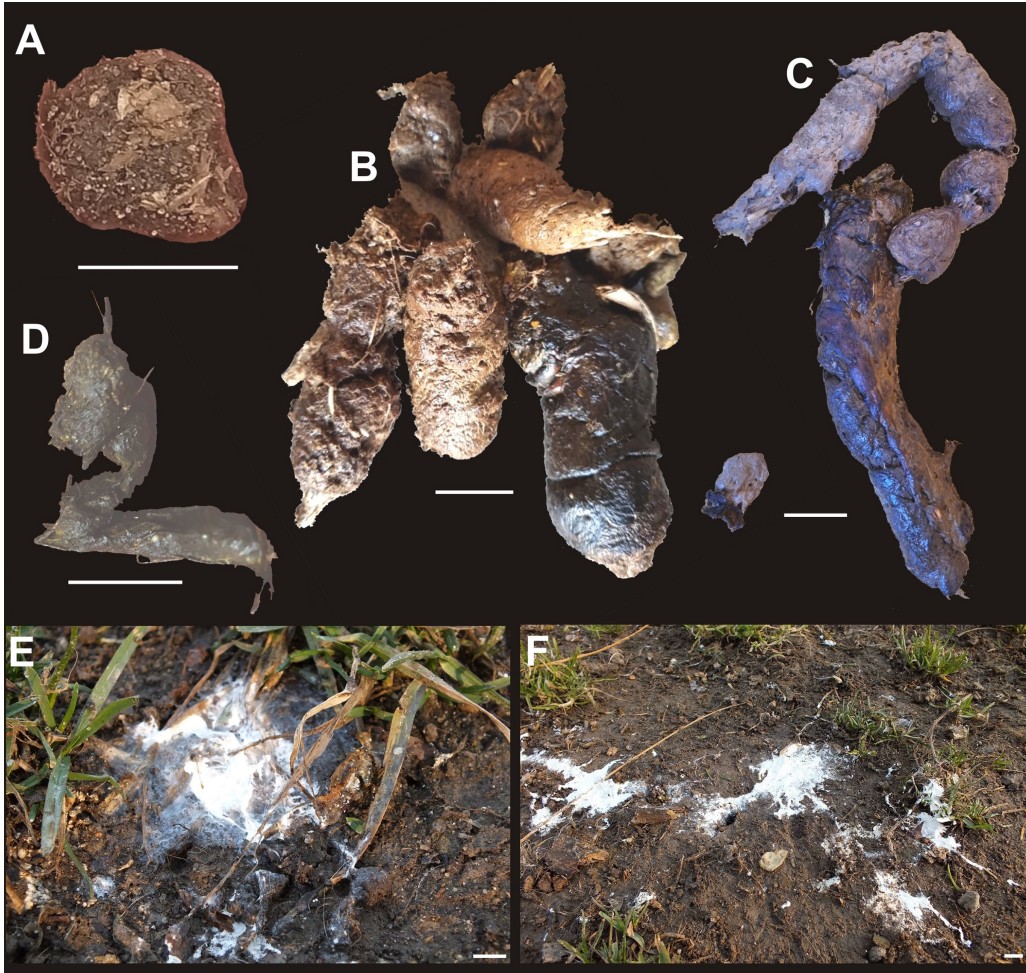

**Figure 10 Examples of modern excrements.** (A) Eurasian beaver (*Castor fiber*). (B) African lion (*Panthera leo*). (C) Cheetah (*Acinonyx*). (D) Indian star tortoise (*Geochelone elegans*). (E, F) White-tailed Eagle (*Haliaeetus albicilla*). Scale bar equals 10 mm.     

Analyzed concretions are preserved as siderite (pure siderite and magnesian-calcian siderite) with major admixture of hematite and minor additions of goethite, pyrite and calcite (Fig. 9).

The more detailed data on microstructure and elemental composition of excrement-shaped masses were collected by utilizing SEM/EDS. We found that the matrix is composed of irregular forms organized in net-system structures (Fig. 5). These forms are bound by thin walls that often consist of several layers (Figs. 5G, 5H). The chemical analyses showed that the walls consist of iron oxides and the interior is likely filled with iron carbonate. Two types of matrix occurring in both morphotypes (M1 and M2) were distinguished. The first one includes smaller (up to 10 µm in diameter) forms with a broad wall of iron oxides and an empty inner part (Figs. 5A–5F). Occasionally, larger forms with carbonate centers (up to 100 µm in diameter) can also be found in the vicinity of the large voids (Fig. 5G). Due to the presence of unfilled forms in the matrix, a distinct porosity pattern can be visible (Figs. 5A1–5F1). In this matrix type, within specimen of M2-type

**Table 4  Basic statistics for excrement-shaped ferruginous masses.**

| Possible producer | Testudinoidea | Serpentes |
|---|---|---|
| N | 7 | 22 |
| Min (mm) | 29 | 30 |
| Max (mm) | 44 | 63 |
| Mean* (mm) | 33.42857 | 46.27273 |
| Std. error | 1.900591 | 2.215824 |
| Variance | 25.28571 | 108.0173 |
| Stand. dev | 5.02849 | 10.39314 |
| Median* (mm) | 32 | 46 |
| 25 prcntil | 30 | 35.75 |
| 75 prcntil | 34 | 56 |
| Skewness | 1.903729 | 0.021165 |
| Kurtosis | 4.125909 | −1.30672 |
| Geom. mean | 33.13972 | 45.12758 |
| Coeff. var | 15.04249 | 22.46061 |

**Note:**
  * Note that the lengths of two types of coprolites are significantly different (both the parametrict-test and non-parametric Mann–Whitney test report a probability of $P << 0.05$ for equality of the means and medians, respectively).

straight or curly elongated structures, which were also observed under optical microscopy, can be found (Figs. 3, 5A–5F). They can occur as thin (10–99 µm) lines with significantly limited porosity (Figs. 3, 5A–5B1, 5C–5F1). In the widest cross sections, some cellular structure is observed (Figs. 3H, 5A–5B1), while in the narrowest cross sections (Figs. 5C–5F1), characteristic scale-like pattern in observed (Figs. 3F, 3G, 5C–5F1). The second type of matrix consists of larger (up to 30 µm in diameter) forms characterized by thinner walls (Figs. 5I, 5J). Their center is always filled with iron carbonate, so there is no distinct porosity, as well as no elongated structures to be found.

To extend the observations and elemental analysis based on SEM, Raman spectra were collected. The data obtained from Raman spectroscopy allows to differentiate two iron oxides within the walls (Figs. 7A, 7B). The spectrum of the first one has bands at 1322, 662, 407, 296, and 227 cm$^{-1}$ (Fig. 6A), which are characteristic for the hematite (*Hanesch, 2009*). The second mineral forms only very thin (<1 µm) layers in the hematite. The main bands of its spectrum are 684, 553, 397, 299, and 242 cm$^{-1}$ (Fig. 5B), which allow identifying this mineral as goethite. The 1322 cm$^{-1}$ band at the goethite spectrum originated from the admixture of the hematite. The spectrum of the carbonate mineral reveals bands connected to the typical vibrations of the $CO_3$ group (Fig. 7C) indicative of siderite.

The Raman spectroscopy was also used to investigate the thin elongated structures in the matrix. There was no variation in the mineralogical composition of these forms in comparison to the matrix. However, during experiments with the 532 nm laser (green), we observed increased fluorescence in the area of the elongated structures with reduced porosity (Fig. 8). This may indicate that although these structures are composed of the same minerals as the matrix, their original chemical composition was different.

## Comparative actualistic studies

For comparative purposes we investigated modern feces produced by a number of vertebrates, comprising all major groups (*i.e.*, fish, amphibians, reptiles, birds, and mammals) (for details see Table 3). We compared our fossil specimens with excrements of extant fish, amphibians, birds and mammals, but they differ in size and shape, and therefore were not subject to further observations (Fig. 10). We noticed, however, that our excrement-shaped ferruginous masses ascribed to the morphotype 1 are very similar to sausage-shaped excrements produced by two testudinid turtle taxa (*i.e.*, the Mediterranean tortoise (*Testudo hermanni*) and the steppe tortoise (*Testudo horsfieldii*)). Their surfaces are mainly smooth and rarely covered with cracks (*e.g.*, Figs. 2C, 2D); the digested plant debris are sometimes visible on their surfaces (Figs. 2C, 2D). They do not reveal any internal layered structure and display bi—or unidirectional bends of 70–120°.

We also compared these fossil excrements with excrements of other extant turtles (another testudinid, *i.e.*, the Indian star tortoise (*Geochelone elegans*), a geoemydid, *i.e.*, the Spanish pond turtle (*Mauremys leprosa*), and a trionychid, *i.e.*, the Nile softshell turtle (*Trionyx triunguis*)), but the feces of the *Trionyx triunguis* differ in size and shape.

On the other hand, our excrement-shaped ferruginous masses ascribed to the morphotype 2 are very similar to more or less rounded to oval, massive excrements produced by three snake taxa (the king python (*Python regius*), the common boa (*Boa constrictor*), and the king cobra (*Ophiophagus hannah*)). Their surfaces are rough, and often contain some remnants of etched hair and feathers (*e.g.*, Figs. 2H, 2I). Moreover, the faeces of the Korean rat snake (*Elaphe anomala*) are also similar to the fossil morphotype 1; however, they differ in size (they are smaller, cf. Fig. 2L). The excrement surfaces of the latter species are covered by some hairs. We also observed three excrements of the common European viper (*Vipera berus*). They are different in shape and size one from another, their surfaces are covered by etched hair (Fig. 2M).

## DISCUSSION

Although in the Biedrzychowice Fm there are numerous inorganic concretions (*Kasiński et al., 2010*; personal observations, MS, 2020), they have a different external morphology (*i.e.*, they do not reveal a characteristic excrement-like shape) and internal structure (*i.e.*, they lack any coalified inclusions or hair-like structures and locally reveal some zoning; Fig. 4) and mineralogy (*e.g.*, they have minor admixtures of pyrite and calcite). Results of our geochemical, mineralogical, petrographic and microtomographical analyses indicate that excrement-shaped masses from Poland mainly consist of siderite and iron oxide rather than phosphate, and rarely contain recognizable food residues, which may indicate abiotic origins of these structures. However, evidence in support of a fecal origin include: (i) the presence of two distinct morphotypes differing in size and shape, (ii) the presence of rare fine striations on the surface of some specimens, and (iii) the presence of hair-like elongated structures or coalified inclusions.

*Spencer (1993)* argued that parallel striations in the pseudocoprolites from the Miocene of southwestern Washington State might have resulted from passage of the material over the grain of the wood. However, fine striations visible on the surface of two specimens from

Poland are more reminiscent of marks left by the anal sphincter because they are not randomly distributed but are located in the pointed end of the specimens (Figs. 2F, 2G). Likewise, they differ from longitudinal parallel striations observed in the specimens ascribed to gut casts of giant terrestrial earthworms (*Broughton, 2017*).

Although mineralogy of the excrement-shaped masses from Poland is not indicative of coprolites, it may be a product of diagenesis (*Broughton, Simpson & Whitaker, 1977*, *1978*; *Broughton, 2017*). Indeed, *Seilacher et al. (2001)* noted that similar excrement-shaped ferruginous masses from the Miocene of southwestern Washington State might have been altered by secondary processes referred to as the 'roll-fronts' of oxidized groundwater (*Goldhaberet, Reynolds & Rye, 1979*; *Harris & King, 1993*), which dissolves calcite and phosphates bones and precipitates ferroan carbonates. The presence of numerous voids and lack of clay minerals within excrement-shaped ferruginous masses may be consistent with this scenario. Diagenetic replacement of phosphates by ferroan minerals, a common process especially in lignite deposits, might have been additionally aided by the fact that the studied excrement-shaped masses were lying at the top of a poorly permeable mud-clay complex that is covered by a well-permeable sand layer. This likely enabled accumulation of ground water at the boundary of these two layers.

Noteworthy, *Francischini, Dentzien-Dias & Schultz (2018)* recently highlighted that the lack of phosphates in the Triassic coprolites from Brasil can be explained by the complete or nearly complete substitution of apatite by calcite, as occurred in some bone remains. Furthermore, lack of phosphates in the coprolites may be in part also explained by the fact they were produced by predominantly herbivorous animals (*Chin & Kirkland, 1998*; *Fiorelli et al., 2013*; *Bajdek et al., 2016*).

Notably, within the morphotype 1 some tiny coalified debris were noted. The morphotype 2, in turn, contains only some elongated thin structures. They are reminiscent of hairs. Their mean size (52 μm) falls well within the range of the hair diameter (4–160 μm) of extant animals (*Mayer, 1952*; *Schneider & Buramuge, 2006*; *Kshirsagar, Singh & Fulari, 2009*). Furthermore, their morphologies, *i.e.*, some cellular structure observed in the widest longitudinal sections (Figs. 2H, 5A–5B1) and characteristic scale-like pattern observed in the narrowest longitudinal sections (Figs. 5C–5F1), are similar to the inner cellular structure of medulla (wide medulla lattice type *sensu Schneider & Buramuge, 2006*, fig. 2) and outer scale-like layers (regular wave pattern *sensu Schneider & Buramuge, 2006*, fig. 4) of the extant and fossil hairs (*e.g.*, fig. 2b, d in *Meng & Wyss, 1997*; figs. 3–9 in *Taru & Backwell, 2013*). If these excrement-shaped masses indeed represent true coprolites, this may indicate that the digestive system of the producer was highly efficient, *i.e.*, it dissolved and absorbed everything but the prey's hair, which were excreted along with feces. Alternatively, some of the illustated structures (*e.g.*, Fig. 5A) might have been also interpreted as burrows produced by parasites (P. Dentzien-Dias, 2022, personal communication). This possibility reinforces that the excrement-shaped masses from Poland may represent coprolites.

Given the sedimentary conditions of the upper part of the Biedrzychowice Fm in the studied area, *i.e.*, subaqueous (vast swamps and rushmarshes) to subaerial environments (forests, braided rivers), and the fact that the boundary between the land zone and swamps

changed depending on the oscillation of the level of waters (*e.g.*, *Bieniewski, 1966*; *Kasiński et al., 2010*), both terrestrial and aqueous vertebrates can be considered as potential coprolite producers. Also a limited redeposition of partly dried/lithified feces by river is likely given the fact that the studied material was found at the boundary between the mud-clay complex and the layer of river sands. Nevertheless, the observed cracks on the coprolite surface, if resulting from drying, strongly point to subaerial conditions.

The presence of two distinct morphotypes, differing in size and shape suggests that they might have been expelled from the two different producers. Indeed, comparative actualistic study of Recent vertebrate faeces shows overall resemblance of the first morphotype (sausage-shaped with rare coalified debris) to excrements of turtles of the group Testudinoidea. This is further supported by the fact that a testudinoid shell fragment was also recovered in the Turów mine. Within Testudinoidea, tortoises (Testudinidae) are terrestrial, while the other two groups that inhabited and still inhabit Europe (Emydidae and Geoemydidae) are aquatic or at least semiaquatic—as such, it seems more probable that the Polish excrements were produced by a terrestrial testudinid. Testudinoids have been already known in the Polish fossil record, however, their earliest occurrence so far was documented in younger strata, *i.e.*, the middle Miocene (MN 6) locality of Nowa Wieś Królewska near Opole (*Wegner, 1913*). That being said, the single shell fragment from Turów represents the earliest testudinoid occurrence from Poland. Nevertheless, testudinoids are already known from early Miocene localities in the vicinity area of northwestern Czech Republic and southeastern Germany (see Table 2). Other turtle lineages, such as chelydrids and trionychids are found in the same vicinity area—among them, the former group is also found in the middle Miocene of Poland (*Joyce, 2016*), while the latter has never been so far identified from that country (*Georgalis & Joyce, 2017*).

On the other hand, the second coprolite morphotype from the Turów mine (morphotype M2) approaches more in its overall morphology the excrements of extant species of snakes. More particularly, there is a high degree of resemblance with large snake species of Constrictores (*sensu Georgalis & Smith, 2020*; *i.e.*, booids and pythonoids). Nevertheless, an overall resemblance of M2 is also apparent with excrements of large caenophidians, such as the elapid *Ophiophagus* and the colubrid *Elaphe*, while, conversely, smaller caenophidian taxa, such as the viperid *Vipera* and the colubrid *Pantherophis*, seem to produce very differently-shaped excrements, which are relatively thin and tightly curled. As such, it is probable that the excrement shape within snake taxa could be somehow size-constrained and does not have a clear taxonomic/phylogenetic value as per its exact affinities. In addition, lizards can be excluded as possible candidate producers for morphotype M2, as excrements of extant large lizard taxa, such as anguids and varanids (which have also an abundant fossil record in the early Miocene of Central Europe) were much differently-shaped. This being said, on the absence of any accompanying skeletal fossil specimen from Turów, we can only infer that the coprolite morphotype M2 was produced by large, but still indeterminate, snakes. After all, large snakes were rather abundant in the Burdigalian of Central Europe, being also rather diverse, pertaining to a number of different lineages (Booidea, Pythonoidea, Colubridae, Natricidae, Elapidae, Viperidae) (see Table 2). It is noteworthy that snakes are known to maintain of a very

acidic pH during digestion and dissolve and absorb everything but the prey's hair (or feathers) and claws, which are excreted along with waste (*Pope et al., 2007*; *Nørgaard et al., 2016*). Among the (semi)aquatic animals that might have lived in this area are also crocodilians or mammals (such as beavers or otters). However, the feces of these animals are also quite different. For instance, excrements of crocodiles, which also have a highly effective digestive system, typically show the unidirectional (rarely bidirectional) bending (ranging from 130° to 150°), the concave termination and layered internal architecture (*Milàn & Hedegaard, 2010*; *Milàn, Skovbjerg Rasmussen & Dybkjær, 2018*). The faeces of beavers are either round or elongated and contain numerous wood chips or sawdust stuck together (Fig. 10A), whereas the faeces of otters, the so-called spraints, are deposited in loose groups and contain numerous indigested debris (*Pagett, 2007*).

## CONCLUSIONS

The excrement-shaped ferruginous masses and the turtle shell fragment from the early Miocene of Turów mine in Poland have been described for the first time. Although different hypotheses were invoked to explain the origins of similar excrement-shaped ferruginous masses, we favour the hypothesis that at least the specimens from Poland represent true coprolites. Evidence in support of a fecal origin of these structures include: (i) the presence of two distinct morphotypes differing in size and shape, (ii) the presence of hair-like structures or coalified inclusions and (iii) the presence of rare fine striations on the surface. If zoological in origin, the first morphotype (sausage-shaped with rare coalified debris) might have been produced by tortoises (Testudinoidea), whereas the second morphotype (rounded to oval-shaped with hair-like structures) might represent fossil feces of snakes (Serpentes).

## ACKNOWLEDGEMENTS

We are particularly grateful to Marek Mitrenga, Director of the Silesian Zoological Garden, for making it possible for us to observe the feces of modern reptiles. Jakub Jagielski, Andrzej Malec and Adriana Strzelczyk from the Silesian Zoological Garden provided help, logistic support and information on the mode of life and diet of reptiles kept in the Silesian Zoological Garden. We would also like to thank the dozens of breeders from Poland and the Czech Republic for acquiring modern research material. Our thanks are also due to Ewa Dąbrowska, who supported us with advice and provided all logistical assistance during the field works, and to retired excavator operator who donated a tortoise shell fragment to the Institute of Earth Sciences of the University of Silesia in Katowice. Eligiusz Szełęg is acknowledged for providing an access to Olympus BX51 polarizing microscope, Tomasz Krzykawski for XRD analysis, and Paweł Bącal for providing some SEM pictures. Comments by three reviewers (George Mustoe, Paula Dentzien-Dias and Thomas Clements) are greatly appreciated.

### Funding

This research project was funded by the National Science Centre, Poland (www.ncn.gov.pl), Grant No. 2019/32/C/NZ4/00150 for Dawid Surmik, and by the Institute of Geological Sciences of University of Wrocław (subvention no. 501 KD76) for Robert Niedźwiedzki and the Ulam Program of the Polish National Agency for Academic Exchange (PPN/ULM/2020/1/00022/U/00001) for Georgios L. Georgalis.

### Grant Disclosures

The following grant information was disclosed by the authors:
National Science Centre: 2019/32/C/NZ4/00150.
Institute of Geological Sciences of University of Wrocław: 501 KD76.
Ulam Program of the Polish National Agency for Academic Exchange: PPN/ULM/2020/1/00022/U/00001.

### Competing Interests

Bartosz J. Płachno is an Academic Editor for PeerJ.

### Author Contributions

- Tomasz Brachaniec conceived and designed the experiments, analyzed the data, prepared figures and/or tables, and approved the final draft.
- Dorota Środek performed the experiments, analyzed the data, prepared figures and/or tables, and approved the final draft.
- Dawid Surmik analyzed the data, authored or reviewed drafts of the article, and approved the final draft.
- Robert Niedźwiedzki performed the experiments, analyzed the data, authored or reviewed drafts of the article, and approved the final draft.
- Georgios L. Georgalis analyzed the data, authored or reviewed drafts of the article, and approved the final draft.
- Bartosz J. Płachno conceived and designed the experiments, analyzed the data, authored or reviewed drafts of the article, and approved the final draft.
- Piotr Duda performed the experiments, analyzed the data, prepared figures and/or tables, and approved the final draft.
- Alexander Lukeneder performed the experiments, analyzed the data, authored or reviewed drafts of the article, and approved the final draft.
- Przemysław Gorzelak performed the experiments, analyzed the data, authored or reviewed drafts of the article, and approved the final draft.
- Mariusz A. Salamon conceived and designed the experiments, analyzed the data, authored or reviewed drafts of the article, and approved the final draft.

### Data Availability

The 3D scan is available in a repository of the University of Silesia in http://hdl.handle.net/20.500.12128/22508.

All raw image slack data is available at reBus: http://hdl.handle.net/20.500.12128/22544.

## Supplemental Information

Supplemental information for this article can be found online at http://dx.doi.org/10.7717/peerj.13652#supplemental-information.

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
