# Peer review of "Comparative actualistic study hints at origins of alleged Miocene coprolites of Poland"

_PeerJ, doi:10.7717/peerj.13652_

## Round 0.1 · original submission · Major Revisions

You provide a thorough analytical analysis and well-written contribution on the controversial “coprolites” of Turów mine in Poland. It would like to see this important work published, but there are still some crucial points which needs to be addressed before publication. The main points being:

Comparative excrement analyses: you focus on comparisons with particular groups of modern excrement producers but not others (crocodilians, mammals). It might be beneficial to have wider comparisons with different groups of excrement producers and taphonomic scenarios on how these terrestrial coprolites ended up in these depositional environments (see reviewers 1 and 2). In this context, I would also be good to state more clearly if you investigate modern excrements for their inclusions.

Concretions: you focus on rare concretions which you interpret as plausible coprolites but make no comparisons with numerous ones which seemingly do not compare with coprolites which makes your interpretations less convincing (compare reviewers 2 and 3). A comparative analysis with concretions which do not resemble coprolites and a preservation scenario explaining this discrepancy would be crucial to make your arguments more convincing.

Trace fossils: tunnel-like structures in figure 5 should be discussed in greater details as they might have been produced by a helminth larva which would further corroborate their identification as coprolites (see reviewer 2).

Please address these as well as all other points raised by reviewers including those in annotated pdfs.

I look forward to obtaining the revised manuscript.

·

Basic reporting

The manuscript is well organized and written in a clearly understandable way. The illustrations are very good.

Experimental design

The research involves to primary aspects: a description of ferruginous extrusions, and comparison with excrements obtained from a wide variety of extant animals, with a goal of determining what ancient creatures may have been responsible of producing fecal masses that later became replaced by iron minerals. This is a very sound experimental design.

Validity of the findings

I believe that the conclusions lack validity, a result of the authors' decision to only consider terrestrial animals for coprolites that are preserved in sediments that were likely deposited in lakebed or stream deposits. I discuss this topic in detail in an attached letter, and in annotations on my edit of the original manuscript.

Additional comments

Review Comments
2/10/22

The manuscript provides a detailed description of ferruginous objects from Miocene strata that are interpreted as being the fossilized excrement having two morphotypes. The smaller objects are inferred to have been made by tortoises, the larger by snakes.

The manuscript has many positive attributes. The writing is clear and well organized, suitable for a wide range of readers, illustrations are excellent, and numerous references are provided. Given the long-standing controversy regarding the excremental versus abiotic origin of ferruginous extrusions (which have wide geographic and chronological distribution), this paper is an important contribution. The authors provide a clear summary of the controversy, where the lack of internal structure and associated body fossils are consistent features of these occurrences. These characteristics suggest an abiotic origin. In contrast, the shapes of these objects closely resemble fecal excrements. I appreciate the fact the authors limit their interpretation to a specific locality, and do not try to generalize their findings in an attempt to present a model that is alleged to apply to other locations where ferruginous extrusions occur.

Although the manuscript has many positive characteristics, there are also some weaknesses that in my view make it unacceptable for publication in its present form. My recommendation is therefore for either acceptance with major revision, or rejection with a suggestion for resubmission. I will leave it to journal editors to make that call.

I am attaching my edit of the manuscript, which includes many comments and suggestions. These mostly involve small details, not major issues. My intent is to be helpful rather than critical, and I admit that as a reviewer I tend to be wordy. I would also like to say that although I have published some papers on the topic of ferruginous “coprolites”, I am open to accept interpretations that are different from my own. I believe that scientific research benefits from a diversity of viewpoints. As a reviewer, my goal is to evaluate manuscripts based on the quality of the presentation and the adequacy of the supporting evidence, without insisting that the conclusions match my own views, or fit traditional paradigms.

In this instance, my primary concern is related to a fundamental inconsistency between geologic evidence and biologic interpretations. In the description of geologic setting, the ferruginous masses (“coprolites”) are reported to have been found in the Biedrzychowice Formation. This unit is reported to contain numerous paleosol layers where plant roots and tree trunks are preserved in situ. This is consistent with the description of the paleoenvironemnt as representing “vast swamps and rush marshes”. Although paleosols are evidence of subaerial conditions, the “coprolites” occur in a clay/mudstone layer, a sediment type that is characteristic of subaqueous conditions.

Herein lies the problem. In studying 787 excrements from extant creatures, the authors chose to consider only terrestrial-dwelling forms. This selection limited the likely choices to tortoises (smaller “coprolite” morphotype, and snakes for the larger morphotype. This interpretation that has major flaws. First, it omits consideration of other creatures that might have lived in the water column overlying the clay/mud layer. This might include reptiles (crocodilians, pond turtles) or mammals (modern forms include beaver, otter, muskrat, etc.) This ecological niche may have been occupied by ancestral equivalents during the Miocene, but it is an environment conducive to habitation. A second problem is that the clay/mudstone deposit is an unlikely habitat for tortoises and snakes. For example, the modern snake excrements all came from taxa whose habitats that include grasslands, scrubland, savannahs, woodlands, or rain forests. None of these environments are likely to produce clay beds, which require the low-energy depositional environment typical of lake beds or slow-moving streams. Tortoises dwell in dry habitats that include grasslands, savannah, and desert, where soils and sediments are typically sandy. They are not likely to be found in quiet-water environments that favor deposition of clay/mudstone.

A further issue is the authors’ utilization of known coprolite-producers based on previous reports. Many of these occurrences are from geologic settings very different from the study area. A further complication is that for virtually all organisms, the fossil record is very incomplete, and new discoveries are being made all the time. A similar issue occurs with the listing of known Miocene fossils from sites in Central Europe. The situation can be summed up with the axiom “Lack of evidence is not evidence of absence”. The list of known fossil taxa from any location is likely to be very incomplete compared to the total taxonomic diversity.

In summary, the authors have provided very detailed mineralogical analysis of the specimens, using a variety of sophisticated methods. I applaud that careful work. The geochemical processes that could cause organic-rich fecal material to become completely replaced by siderite defy easy explanation (the roll-front replacement hypothesis cited in the text came from a paleontologist who had no credibility as a geochemist). This is an attraction of the paper: it adds new observations to a controversial topic. However, the contention that the objects originated as excrements from tortoises and snakes is not supported by geologic or paleocologic evidence. I see this as a fatal flaw for the acceptance of a manuscript that is otherwise an important contribution.

I welcome having my identity revealed to the authors.

George Mustoe
Research Associate
Geology Department
Western Washington University
Bellingham, WA 98225 USA.

·

Basic reporting

The manuscript is well written and contribute to the knowledge of coprolites, especially those that can be mistaken with concretions. However, there are important reviews to be done before it is ready for publication:

- Some words are attached to each other and are marked in the PDF;
- Some important papers can improve the discussion (such as Francischini et al., 2018; Milán, 2012).

Experimental design

The methods used are sound, the inclusions are described in good detail, and the figures are clear and easy to interpret. However some images can improve the paper:
- It is not clear if the inclusions of the modern excrements were analyzed, make it clear;
- Provide images of the concretions (morphologies and the internal structure) and show the diferences with coprolites;
- The authors analyzed the excrement morphology of modern fish, amphibians, reptiles, birds and mammals. A small sample of the images taken of this variety of excrements can be added as a supplementary material.

Validity of the findings

- Usually, siderite and iron oxide masses are considered concretions, the authors say that in the Biedrzychowice Fm have numerous concretions with the same chemical composition of the coprolites. They must provide a new figure showing the differences between the concretions and coprolites;
- The absence of inclusions must be better discussed. Crocodiles have a very effective digestive system, and are found in the Miocene deposits, while the serpents faeces described by the authors are not clear if there were bones;
- The tunnels shown in figure 5 can be better discussed, specially the 5a. It resembles perfectly a tunnel produced by an invertebrate. Due the size, could be produced by a first stage larva of a helminth. This possibility reinforces that these structures are truly coprolites. The authors should discuss this.

Additional comments

See the PDF for more comments.

·

Basic reporting

Reviewer comments:
Thank you for asking me to review this interesting paper! It covers a very interesting subject matter that seems quite difficult to unpick and the authors have made a good (and convincing) attempt to understand these potential fossil poops!
The figures are well made and easy to interpret. The paper seems well referenced (although please note I am not a coprologist). The methods are well described and appropriate – using an interesting method to do some comparative morphological work with extant animal faeces, and I think the findings shown in this paper are sound and sensible. The paper is fairly easy to read, although I think that some minor revisions to the abstract and introduction would greatly help the reader. I would have also liked a little more investigation into the potential diagenetic mode of preservation for faeces to become siderite, but I do understand that maybe outside of the scope of the paper.
I have one concern pertaining an aspect of the paper which I did not understand well. This maybe my fault, but I was not clear whether there are lots of siderite concretions found in this deposit which do not look like the morphotypes identified by the authors (see my comments RE: line 385). This is concerning and I would like to see how this addressed, because, at the moment it undermines the findings of the paper.
If this can be addressed then I would recommend this fun and fascinating paper for publication.
I have made a few detailed comments in my PDF attached.
All the best,
Dr Thomas Clements

Experimental design

No comment

Validity of the findings

No comment

---

## Round 0.2 · Minor Revisions

Thank you for thoroughly addressing the raised point and suggestions. There are some minor additional points i would like you to implement before publication (see reviewer 1). Thank you for also providing the raw data and a video of the model but please also consider providing the final 3D-model as an .STL file - compare Davies et al. 2017: https://doi.org/10.1098/rspb.2017.0194

·

Basic reporting

The manuscript is well-written, with clear organization and good illustrations. Conclusions are based on abundant data.

Experimental design

The comparison of suspected coprolites with excrements from living organisms is a well-designed strategy.

Validity of the findings

The authors do a good job of presenting evidence that supports their hypothesis. Personally, I am skeptical of the interpretation of the coexistence of snakes and tortoises as the only producers of coprolites (it's an unusual ecological assemblage), However, the origin of ferruginous "coprolite" extrusions has been a subject of much controversy, and I believe that this new study is a useful contribution.

Additional comments

The authors have paid careful attention to the comments from reviewers, and I recommend the manuscript for acceptance pending some minor changes. These are small issues that would likely be detected and remedied during final production editing. Here are some examples:
Introduction, paragraph 1: change spelling of "undegested" to "undigested".
Materals and Methods, paragraph 2: The description of screening of sediments concludes with "Unfortunately nothing was found..,." This introduces a subjective judgement to the analysis. A better statement would be to delete "unfortunately" and just say "no fossils were found".
Mineralogy, geochemistry and structural analysis, paragraph 3. This present wording suffers from a dangling participle kind of problem caused bu the vague use of "it". : "The spectrum of the carbonate...... It can be recognized as siderite." This wording is saying that the spectrum is made of siderite.
Discussion, paragraph 4. Fix spelling for "communitaction"
Also, a bit later, "afterall" should be "after all).
The last section of the Discussion uses the spellings "feces" and faeces" in a single sentence.

These are very minor issues that will be easy to correct.

---

## Round 0.3 · accepted · Accept

Thank you for addressing these final typographic and formatting suggestions as well as adding an .stl file. Your manuscript is now good to go and I look forward to seeing it published.